# Effectiveness of Anodal otDCS Following with Anodal tDCS Rather than tDCS Alone for Increasing of Relative Power of Intrinsic Matched EEG Bands in Rat Brains

**DOI:** 10.3390/brainsci13010072

**Published:** 2022-12-30

**Authors:** Nafe M. Al-Tawarah, Zulal Kaptan, Hashem A. Abu-Harirah, Mohammad Nofal, Belal Almajali, Sultan Jarrar, Zubeyir Bayraktaroğlu, Haitham Qaralleh, Khaled M. Khleifat, Ziya Y. Ziylan, Rawand H. Al dmour, Moath Alqaraleh, Sacit Karamursel

**Affiliations:** 1Department of Medical Laboratory Sciences, Faculty of Science, Mutah University, Mutah 61710, Jordan; 2Department of Physiology, Faculty of Medicine, Beykent University, Bolu 14030, Turkey; 3Medical Laboratory, Faculty of Allied Medical Sciences, Zarqa University, Zarqa 13110, Jordan; 4Department of Surgery, Faculty of Medicine, Mutah University, Kerak 61710, Jordan; 5Neurosurgery Department, Faculty of Medicine, Jordan University of Science and Technology, Irbid 22110, Jordan; 6Department of Physiology, Istanbul Faculty of Medicine, Istanbul University, Istanbul 34093, Turkey; 7Pharmacological and Diagnostic Research Center (PDRC), Faculty of Pharmacy, Al-Ahliyya Amman University, Amman 19328, Jordan; 8Department of Physiology, Faculty of Medicine, Koc University, Istanbul 34450, Turkey

**Keywords:** tDCS, otDCS, EEG, frontal cortex, relative band power

## Abstract

Background: This study sought to determine whether (1) evidence is available of interactions between anodal tDCS and oscillated tDCS stimulation patterns to increase the power of endogenous brain oscillations and (2) the frequency matching the applied anodal otDCS’s frequency and the brain’s dominant intrinsic frequency influence power shifting during stimulation pattern sessions by both anodal DCS and anodal oscillated DCS. Method: Rats received different anodal tDCS and otDCS stimulation patterns using 8.5 Hz and 13 Hz state-related dominant intrinsic frequencies of anodal otDCS. The rats were divided into groups with specific stimulation patterns: group A: tDCS–otDCS (8.5 Hz)–otDCS (13 Hz); group B: otDCS (8.5 Hz)–tDCS–otDCS (13 Hz); group C: otDCS (13 Hz)–tDCS–otDCS (8.5 Hz). Acute relative power changes (i.e., following 10 min stimulation sessions) in six frequency bands—delta (1.5–4 Hz), theta (4–7 Hz), alpha-1 (7–10 Hz), alpha-2 (10–12 Hz), beta-1 (12–15 Hz) and beta-2 (15–20 Hz)—were compared using three factors and repeated ANOVA measurement. Results: For each stimulation, tDCS increased theta power band and, above bands alpha and beta, a drop in delta power was observed. Anodal otDCS had a mild increasing power effect in both matched intrinsic and delta bands. In group pattern stimulations, increased power of endogenous frequencies matched exogenous otDCS frequencies—8.5 Hz or 13 Hz—with more potent effects in upper bands. The power was markedly more potent with the otDCS–tDCS stimulation pattern than the tDCS–otDCS pattern. Significance: The findings suggest that the otDCS–tDCS pattern stimulation increased the power in matched intrinsic oscillations and, significantly, in the above bands in an ascending order. We provide evidence for the successful corporation between otDCS (as frequency-matched guidance) and tDCS (as a power generator) rather than tDCS alone when stimulating a desired brain intrinsic band (herein, tES specificity).

## 1. Introduction and Aim

Transcranial direct current stimulation (tDCS) is a non-invasive, painless, and easily applied technique, used for the alteration of brain stimulation level by modulation of cortical excitability through non-synaptic mechanisms [1]. This technique has been successfully applied to the human cortex and has increasingly gained interest among clinical neuroscientists [2]. Although tDCS has not yet been proven to be an effective therapy for all brain disorders, it has already been used for many neurological and psychiatric disorders, including chronic stroke [3], epilepsy, chronic pain [4], and Parkinson’s disease [5]. These efforts were made aiming to improve the efficiency of tDCS’ applications in human subjects, in both healthy cases and clinical ones [6].

Oscillatory brain activities are changes of extracellular electric field evoked by neuronal network activities that can be observed using the electroencephalograph (EEG) [7]. Clinically, EEG studies have exhibited abnormal brain rhythms in many neuropathological cases, such as Parkinson’s disease, Alzheimer’s, and epileptic seizures [8,9,10]. Many previous studies pointed to the quantification role of these EEG bands for assessment of some ailments and physiological biomarkers [11,12]. Studies have found that the electromodulation of the ongoing rhythms of the brains of these patients has potential effects on the cognitive process—targeting specific brain oscillations—and all this can be related to otDCS [13]. Unlike tDCS, the application of anodal-otDCS (i.e., current is oscillated around a certain positive DC value) mimics these endogenous oscillations of brain electrical activity [14].

During transcranial electrical stimulation (TES) studies, using animal models in parallel with other medical research studies provides a powerful tool to study the many aspects of TES and perform a standard stimulation protocol for TES application in humans [15]. Similar to human beings, the prefrontal cortex (PFC) is the main targeted area in the rat’s brain for its implications on cognitive functions, working memory, attention, and learning [16]. The electrostimulation of the rat’s frontal cortex (FC) leads to valuable EEG changes that were investigated for the assessment of TES after-effects [17]. Differing from skin electrodes, used in humans during tDCS, the fixation of the electrode in the rats will be cranial or transcranial to ensure contact position and define the contact area of the electrode over the targeted brain cortex [18]. According to their protocol, ref. [19] demonstrated that anodal-tDCS at a current strength of 200 μA for 20 min was capable of inducing a localized increase in cortical stimulation depression (CSD) velocity in the rat cortex without any neural lesions. They used a standard screw with a defined contact area over the rat skull of approximately 3.5 mm^2^.

It is well known that the modulation of brain oscillations is a frequency-specific electrical stimulation manner; thus, for this study, the neuromodulation effect of oscillated TES was made a priority rather than direct current TES [20]. Therefore, we hypothesized that coupling otDCS with anodal tDCS might produce significant changes in the brain’s electrical activities and excitability level. For that, this work focuses on studying all the acute changes in the EEG of underlying brain oscillations, induced by different stimulation patterns of both otDCS and anodal tDCS. Accordingly, the present study was designed to determine whether (1) evidence is available for interactions between well-known anodal tDCS and oscillated tDCS stimulation to increase the relative power of endogenous brain waves more than can be achieved by tDCS alone and (2) to investigate if the frequency matching between applied frequency of anodal otDCS’s and the brain’s state-related frequency influences power shifting during different stimulation patterns.

## 2. Materials and Methods

### 2.1. Animals

Twenty-four Wister albino rats (Experimental Medicine Research Center of Istanbul University), aged 10–12 weeks old (weight 240 ± 30 gm: mean ± SEM) at time of surgery, were investigated during this study. Animals were housed individually with ad libitum access to water and food. In addition, all rats were housed in standard conditions; normal light/dark cycles (light12 h), light from 07.00 am to 7.00 pm. In all the stages of this work, rats have been used and handled in proper way that met all the requirements and instructions of Laboratory Animals Ethics Committee of Istanbul University (decision No. 46/2014). In which, all efforts were made throughout the surgery, stimulation, and EEG-recording for using minimum number of animals and to avoid suffering of animals.

### 2.2. Surgery

As demonstrated by Liebetanz et al. [21], rats were anesthetized with ketamine hydrochloride (50–70 mg/kg i.p.) and xylazine (6–8 mg/kg i.p.) (Alfasan International B.V., Holland). In fact, according to Abdullah and Islam [21], additional ketamine (5 mg/kg i.p.) should be given during the surgery by squeezing the footpad in case of any appearance or sign of pain stimulus. Additionally, a combination of lidocaine (0.5%) with epinephrine (1:200,000) (ADEKA, Istanbul) was injected subcutaneously along the incision site (maximum 0.1–0.2 mL) as a local anesthetic and vasoconstrictive agent, respectively [22]. Finally, approximately 0.04 mg/kg atropine (0.2%) (DEVA, Topkapi, Istanbul) was administered subcutaneously as a parasympatholytic agent to aid breathing [23].

A surgical procedure was conducted for chronic implantation of transcranial and epicranial electrodes for both tDCS/otDCS, as well as for EEG recording. The surgical procedure was carried out in accordance with many studies [19,24,25,26,27,28] with some modifications according to the study requirements. After transferring a rat to the stereotaxic apparatus (Technology Support, RWD Life Science Co., Ltd., Guangdong, China), the rat’s head was fixed in a proper flat position. After the preparation of the surgical site, a mid-sagittal incision of approximately 2–3 cm was made, followed by a reflection of the skin with hemostasis to expose the entire dorsal portion of the skull. The skull surface was cleaned and dried with hydrogen peroxide (10%) (Zhengzhou Sino Chemical Co., Ltd., Guangdong, China). By using a micro drill (Technology Support, RWD Life Science Co., Ltd., Guangdong, China) with a diameter head of 1 mm, six holes were made with specific coordinates (Figure 1). Initially, the bregma was marked as a reference point for the determination of all needed coordinates. From one side, for bilateral electrical stimulation of FC, two holes were made transcranially with the following coordinates: AP: +3.9 mm, L: ±2 mm, and DV approximately −0.5 mm. From another side, for epidural EEG recording, two bilateral holes were made over FC through the skull: AP: +1.6 mm, ML: ±2 mm. In addition, another anterior hole (AP: +6.9, L: +1.1) was made as a ground electrode. The final hole was made over the cerebellum (AP: 12.0 mm, ML: ±0 mm) to be used as a return electrode for bilateral electrical stimulation and as a reference electrode for epidural EEG.

The investigated electrodes in this study were screws made from stainless steel with a diameter of 1.2 mm and a shaft length of 2.5 mm (Aliexpress, San Mateo, CA, USA). During the surgery, sterilized screws were fixed over the rats’ skulls in certain positions as tightly as possible. All electrodes were connected to a five-pin gold plated male-connector with properly isolated micro-wires. All implanted screws, micro-wires, and the male connector were fixed tightly to the skull surface by hump-shaped dental cement (zinc phosphate cement; Dis Malzemeleri Sanayı Tıcaret AŞ., Istanbul, Turkey) in which just the five male pins appeared dorsally (Figure 2). In fact, after filling all incision gabs with dental cement, the loose appearance of any part of the incision edge was sutured with proper rat surgical sutures 4-0 or 3-0 (Propilen non-absorbable monofilament, Dogsan, Trabzon, Turkey).

After the surgery, local and systematic antibiotics (e.g., Bacitracin and Gentamicin, respectively) were used once a day for three successive days to avoid post-operative bacterial infection. For postsurgical recovery, the rats were housed in cages placed inside a warm and clean room for up to five days, during which their health was monitored, and postoperative care performed daily. More than ten rats were prepared surgically for each investigated group (*n* = 8). After several postoperative days (7 to 10 days), for each group, just eight rats from the total ten surgically implanted rats that exhibited both normal health conditions and normal behavior were accepted in the study. In fact, rats that did not exhibit normal behavior or had poor health conditions had been excluded from the study completely.

### 2.3. Stimulation Parameters for Both Anodal tDCS and otDCS

The stimulation parameters, including current intensity, duration of stimulation, and the range of used oscillations, were determined prior to the formulation of the stimulation protocol.

### 2.4. Current Intensity and Time Duration

Dockery et al. [2] demonstrated that the application of 0.2–0.5 mA anodal tDCS over the PFC of rats leads to short-term as well as long-term improvements of cognitive function, elaborated by the changing of the PFC’s excitability. During their research on the rat cortex, to study some aspects related to tDCS, Rueger et al. [29] and Fregni et al. [30] used a current intensity of 0.2–0.5 mA and 0.2 mA/20 min, respectively, in their TES protocol. Moreover, Liebetanz and his colleagues (2009) determined the model stimulation parameters that prevent brain injury [29]. Therefore, in our study, we adopted tDCS/otDCS parameters in accordance with the parameters recommended in previous studies. Therefore, the contacting surface area of the used anodal electrode was approximately 3.5 mm^2^, the applied current intensity approximately 200 μA with the duration of stimulation being 15–20 min [31].

### 2.5. Oscillation Values

It was found that the habituation of the rats in a restricted area to remain quiet was essential to decrease artifacts of movement during EEG recording [32]. For this, the rats were placed inside a closed box that was prepared for this purpose; it was approximately 20 cm in diameter and 40 cm in height. In fact, the box was electromagnetically isolated by aluminum foils and kept away from sound and noise sources as much as possible. Thus, the state-endogenous brain oscillations of our rats under these conditions were determined before starting our serial experiments. The rats had been habituated for three successive days in these conditions before the study. Additionally, before the onset of the stimulation protocol and under the study’s prepared conditions, the EEG records from some rats had been harvested. The harvested EEG records were investigated to determine normal exhibited oscillated values (dominant frequencies) that should be adopted through the otDCS. All harvested normal EEG records were transferred to LabScribe 3 in which spectral analysis was performed using a fast Fourier transform (FFT). After this, the main exhibited dominant endogenous frequencies of the rat’s FC were obtained to be investigated later in the study. In fact, two main dominant frequencies were adopted during this study: one was nearly at the borderline between the theta and alpha bands: 8.5 Hz; the other one was between the alpha and beta bands: 13 Hz.

### 2.6. Stimulation Protocol and Experimental Design

Our surgically prepared animals (*n* = 24) were randomly divided into three groups of stimulation patterns (A, B, and C). The TES/EEG recording protocol was initiated by moving the rat with an implanted male five-pin connector inside the recording box, followed directly by connecting the male connector with a female five-pin connector. The five-pin female connector was connected with an EEG system and a brain electro-stimulator device (Teknofil, Istanbul, Turkey), capable of stimulating DC or oscillated DC with 0.1–10 mA and 0.1–100 Hz (Figure 3).

The timeline plan protocol of the TES-EEG recording series for each group’s stimulation pattern was conducted as shown in Figure 4. All EEG data were recorded using the iWorx LabScribe1.7 software (iWorx System Inc., Dover, NH 03820, USA). in which the preferences (output display format) were adjusted as follows: Two channels A and B recorded EEG signals from right and left FC, respectively. Data were continuously recorded pre- and post-stimulation for 5 min. The bandpass filter was between 0.3 Hz high-pass filter and 50 Hz low-pass filter, the sample frequency was set at 200 Hz. The display time 10 s/page, and finally, the Ymax and Ymin voltage scale were 5 V and −5 V, respectively.

### 2.7. EEG Data Processing and Analysis

From each side of FC, harvested EEG records (right and left) were stored and processed separately and independently to fulfill the aims of this study. At least 5 min of recorded EEG were collected in each recording time period. After recording all the EEG data using the iWorx LabScribe 1.7 system, all harvested EEG data were stored in the working computer. As the iWorx LabScribe1.7 system is not capable of analyzing such types of data, all stored EEG data were, therefore, transferred to the iWorx LabScribe3 system. During the study, EEG recording was designed to decrease noise/artifact as much as possible. EEG recording may be contaminated with external electrical artifacts which result from EEG amplifier, tDCS stimulator, and cables. For that, animal was placed in constricted area electrically isolated by aluminum foils. In addition, at the time of stimulation, the EEG system was turned off. About the intrinsic noise/artifact like cardiac/muscle movement/eye blinking/jaw related artifacts, limited place helped to decrease as many artifacts as possible. Two programs were used to recognize and eliminate noise/artifact signals: LabScribe (iWorx System Inc., Dover, NH 03820, USA) and MATLAB (Version 7.13.0.561 (R2011b), The MathWorks, Natick, MA, USA, 2011) [3]. In LabScribe, it gives a feature of manual removal of unusual signals, as well as by setting the system on the cutoff frequencies 0.3 Hz (high pass) and 50 Hz (low pass) filters. Finally, data have been processed and analyzed using EEGLAB (Running in MATLAB). Independent Component Analysis (ICA) algorithm in EEGLAB has a capability to removed noise/artifact signals. EEG signals were analyzed off-line set using FFT. In fact, the channel (left or right) that showed fewer artifacts and less noise with enough 300 s EEG recording was adopted in this study for each group. Finally, each artifact-free EEG record was processed by MATLAB to obtain the power spectral values for the range of frequencies (1–30 Hz) in a 0.5 Hz step. The EEG spectral power changes were investigated and analyzed using the relative band power (RBP), for the following six bands: delta (1.5–4 Hz), theta (4–7 Hz), alpha-1 (7–10 Hz), alpha-2 (10–12 Hz), beta -1 (12–15 Hz), and beta-2 (15–30 Hz), in which the value of RBP for each corresponding band represents the percentage of absolute power of such band to overall captured bands.

### 2.8. Data Statistical Analysis

The harvested RBPs of all pattern groups were analyzed using SPSS version 22 (SPSS Inc., Chicago, IL, USA) in which *p* < 0.05 was considered as a significant result. On one hand, six tables of all investigated bands showed pre- and post-stimulation RBPs for all stimulations within all three-group patterns. For each investigated band, repeated measurements using the SPSS (three-factor repeated measurement ANOVA; tDCS, 8.5otDCS, and 13otDCS) were conducted to compare the mean differences between all pre-stimulation and all post-stimulation RBPs and, consequently, to find if there are any significant results between these groups. Then, a one-way ANOVA was conducted to find if there were any significant differences between the baseline level RBP before initiation of the stimulation pattern and the RBP of the last acute after-effect in each group. All significant results were shown by the ANOVA test, and this was followed by post hoc tests (Tukey’s methods) to find pairs that were significantly different.

Inter-individual differences were present in all investigated rats, as well as differences between the base-line level of the whole stimulation pattern and after each one, within the stimulation pattern. Therefore, for each band, the power differences in the RBPs for all stimulations within stimulation patterns were determined. The power difference (dP) of each anodal stimulation (tDCS and otDCS) within the stimulation pattern was measured for all bands according to the following formula: dP: P1 − P0, where P1 denotes the acute after-effects of the RBP and P0 denotes the pre-stimulation RBPs. After that, a one-way ANOVA was conducted for each investigated band to determine whether there were any significant differences between all mean dPs within the same group pattern for each band (e.g., in Group A: mean dP by tDCS, cumulative mean dP by 8.5otDCS, and cumulative mean dP by 13otDCS). In addition, a one-way ANOVA was conducted between the mean dPs of the same stimulation sessions within all three groups for each band (e.g., first stimulation session: A group, tDCS; B group, A8.5; C group, A13). Finally, all significant results shown by the ANOVA tests were followed by post hoc tests (Tukey’s methods) to determine the pairs that were significantly different.

## 3. Results

One-way ANOVA was also conducted between the normal (baseline) RBP’s level and the last acute RBP’s level (stimulation pattern effect) for each stimulation pattern. Figure 5 and Figure 6 (mean changes of RBPs in Figure 5) show some significant results in power band changes corresponding to each group pattern (this has been pointed out with a small letter after the band). The results show drops in deltaA power (*p* < 0.05; F (2, 14)3.7 = 4.2) and an increase in the power of these bands: alpha-1A, alpha-2A, alpha-2B, beta-1A, beta-1B, and beta-2B (*p* < 0.05, F (2, 14)3.7 = 4.7, 5.7, 21, 5.2, 10.2, and 4.2, respectively).

The results revealed some valuable significant notes, such as the fact that both stimulation pattern groups, A and B, exhibited a significant decrease in delta power, with a more potent dropping effect exhibited by the Group A stimulation pattern than the Group B one, while in Group C, there was no effect at all. It appeared that the Group A stimulation patterns (started with tDCS) increased the power of the theta band and those above in different non-significant ranges. In addition, one can note that after the stimulation with Group B patterns, the relative power of the alpha-1B band increased more potently than in the thetaB and deltaB bands. Additionally, Group B markedly increased the relative power of the above bands (alpha-2B, beta-1B, and beta-2B) in an ascending manner. Moreover, for the Group C pattern, one can note that the increase in relative power started with beta-1C and continued in beta-2C.

The cumulative power differences (dP) in each stimulation pattern were investigated for further analytical purposes by using one-way ANOVA initially, for assessment of any significant changes in power differences due to each stimulation within each stimulation pattern cumulatively (tDCS alone and then cumulatively by 8.5otDCS and 13otDCS). The results revealed that significant changes in dP had been induced by Group A’s stimulation pattern in the following four bands: deltaA, thetaA, alpha-1A, and alpha-2A (*p* < 0.05 and F (2, 10)4.1 = 4.76, 4.2, 3.5, and 3.8, respectively) (Figure 7). However, no significant changes in dP were revealed in beta-1A and beta-2A bands at all. In the delta band, the results showed a marked significant decrease in the band power after tDCS stimulation (approximately −19%). The post hoc test showed a significantly smaller dropping effect on dP by 8.5otDCS and much more by 13otDCS, compared with tDCS (*p* < 0.05, MD = −0.1619 and 0.1899, respectively). In the theta band, the results showed an increase in the band power after tDCS stimulation (by approximately 6%). The post hoc test showed a significant drop in dP after 8otDCS and 13otDCS, compared with tDCS (*p* < 0.05, MD = 0.065 and 0.061, respectively).

In the alpha-1 band (Figure 7), the post hoc Tukey’ tests for multiple comparisons between tDCS and 13otDCS effect (*p* = 0.06 and 0.05, respectively, MD = 0.061 and ±SEM = 0.023) showed a marked increase in alpha-1 power followed by a marked decrease in dP level by 13otDCS. Additionally, the mean dP decreased by 8.5otDCS, compared with tDCS (MD = −0.043 and ±SEM = 0.023). Finally, in the alpha-2 band (Figure 7), the post hoc Tukey’s tests for multiple comparisons revealed a marked increase in mean dP by tDCS followed by a drop in 8.5otDCS (*p* = 0.053 and 0.044, respectively, and MD = 0.026). Furthermore, a mild increase in the mean dP appeared after stimulation with 13otDCS compared with 58.5otDCS (MD = −0.006 and ±SEM = 0.025). Thus, Group A pattern results showed that tDCS made a significant decrease in delta band power and increases on the other upper bands with different levels. On the other hand, 8otDCS made a mild decrease in both delta and theta band power and, in an ascending manner, made a significant increase in the power of both alpha-1 and alpha-2. Finally, 13otDCS seemed to make no effect on delta power, a drop in both theta and alpha-1 bands, and a mild increase in the power of the alpha-2 band. These results concur with previous findings (changes in RBPs between last acute and baseline EEG recordings) very well.

In addition, a one-way ANOVA was conducted for the mean dP values of the same stimulation session within all group patterns. Results showed that first stimulation session in each pattern, rather than the second and third stimulation session, showed significant changes in dP in the following bands: delta, theta, alpha-1, alpha-2, and beta-1 (*p* < 0.05, except alpha-1: *p* = 0.09, F (2, 12)3.8: 3.99, 4.9, 2.71, 4.67, and 4.7, respectively) (Figure 8). In this, tDCS (first session in Group A) produced a decrease in dP in the delta band, while both 8.5otDCS (first session in Group B) and 13otDCS (first session in Group C) made a significant increase in the mean dP of the delta band. A post hoc test revealed that 13otDCS has more potent power to increase the power of the delta band than 8.5otDCS. In the theta band, the post hoc test showed that tDCS produces a significant increase in the theta band power, while 8.5otDCS and 13otDCS cause a decrease in theta power compared with tDCS (*p* < 0.05, MD: 0.116 and MD: 0.104, respectively).

In the alpha-1 band (Figure 8), the results show that tDCS increases the mean dP of alpha-1 (*p* = 0.21, ±SEM: 0.037), while 8.5otDCS and 13otDCS make a mild decrease in the mean dP, compared with tDCS (MD: 0.065 and 0.087, respectively, ±SEM: 0.037). In addition, results show that tDCS produced a significant increase in the mean dP of the alpha-2 band in comparison with the tDCS effect on the alpha-2 band, which, as per the post hoc test, shows a significant decrease in dP by 13otDCS and a mild elevation in dP by 8.5otDCS (MD: MD: 0.036 and 0.17, respectively, ±SEM: 0.011). Finally, the results show that tDCS produced a significant increase in the mean dP of the beta-1 band (Figure 8). However, in comparison with the tDCS effect, the post hoc test showed a mild decrease in the dP of the beta-2 band by the 13otDCS and a mild elevation by the 8.5otDCS (MD: 0.031 and 0.17, respectively, ±SEM: 0.011). Notably, a single effect of the first stimulation session in all stimulation patterns showed some significant results. Group A tDCS decreased the power of the delta band and gradually, in a descending manner, increased the power of the upper bands (alpha-1, alpha-2, and beta). In Group B, the 8.5otDCS increased the power of the delta, alpha-2, and beta-1 bands while decreasing the dP of both theta and alpha-1. Finally, in Group C, it was found that 13otDCS increased the power of the delta band while decreasing the dP of the theta band, in a descending manner, up to the beta-1 band, in which the dP of the beta-1 band reached near-zero levels.

## 4. Discussion

At the beginning of this study, we supposed that the stimulation of the PFC of rats with different patterns of anodal oscillated tDCS/anodal tDCS would entrain or synchronize targeted brain oscillations. To achieve this goal, during the otDCS, the applied exogenous frequencies should be matched with the normal intrinsic dominant brain frequencies (i.e., the main peaks that appear during the spectral analysis of normal EEG signals that represent the most-exhibited state-related brain oscillations). Many studies suggested that entrainment of intrinsic brain oscillations with external oscillated TES will be more pronounced when the externally applied frequency matches with the dominant exhibited endogenous oscillation (frequency tuning) [6,33,34]. Precisely, these studies revealed a closed relation between the clinical symptoms and those exhibited endogenous oscillations (Table 1). Therefore, all efforts concerning brain stimulation were oriented to increase the performance and effectiveness of this technique for the enhancement of human cognition and the treatment of neurological disorders [35]. The results of the present study may add and complement such studies aiming for the improvement of the brain stimulation protocol. It consequently might create desired effects during transcranial brain stimulation and, thus, increase tDCS performance.

During this study, the harvested mean EEG RBPs from each pre- and post-stimulation (in three stimulation patterns (A, B, and C) were compared within the subject variables by using repeated-measures ANOVA (three variables: tDCS, 8.5otDCS, and 13otDCS). All significant results and groups were compared using t-tests and corrected for multiple compressions. The results showed no significant changes in all pre- and post-stimulation RBPs among all investigated bands, except for within post-stimulation variables of beta-1 (12–15 Hz). This significant result showed a marked increase in the RBP of the 12–15 Hz intrinsic band after stimulation with 13otDCS and, more increasingly, after stimulation with tDCS. Although some studies suggested that frequency-matching between exogenous oscillations and endogenous ones leads to better neuromodulation results, compared with non-matching frequency stimulation. However, most RBP findings (pre- and post-stimulation) revealed no significant results. This might hypothetically be related to inadequate TES parameters (e.g., duration, intensity, etc.) for modulating intrinsic oscillations and, thus, for making sufficient power changes [20,36]. In fact, our novel study had far different targeted aims, primarily to reveal whether there was any evidence of a specific interaction between tDCS and otDCS stimulation patterns to increase the power of desired endogenous oscillations and to study whether the frequency matching between the otDCS and the targeted dominant intrinsic frequency plays a role in the shifting of power. In fact, the preliminary findings were not enough to confirm these goals; thus, further analysis was performed to search for evidence of their existence.

A one-way ANOVA revealed some of the novel significant results in mean RBPs between the acute last stimulation and the baseline level for each stimulation pattern group (A, B, and C). It was clearly apparent that all stimulation patterns had negative power effects on the delta band power, which was more potent in Group A (Delta-A) (Figure 5). If we compared previous results with Figure 7 and Figure 8, one can conclude that during Group A’s pattern, tDCS was responsible for dropping delta power, while otDCS increased it mainly by 13 Hz. This result corresponded with the findings of previous studies that anodal tDCS causes a significant dropping effect of delta band power on PFC [37,38,39] and cathodal tDCS causes a reverse effect [40,41]. In addition, one study revealed that oscillated tDCS increases the EEG power of the delta band beneath electrode sites [42]. In fact, although externally applied frequencies via otDCS did not coincide with the delta band, a mild increase in the delta power was observed by otDCS at 8.5 Hz and with more potency by otDCS at 13 Hz. This might be related to the non-specific effects of otDCS that lead to a significant increase in the power of some non-matching endogenous oscillation [43,44].

Furthermore, the results showed that the Group A pattern produced a marked increase in all bands above the delta band on different levels (Figure 5). By connecting this result with those of the post-stimulation Group A pattern (Figure 8), the findings showed that tDCS alone has a sufficient general power effect for increasing the power of those bands, without otDCS. In Group B, the results revealed that otDCS has a guidance-like effect with a mild increase in the power of analogous intrinsic bands (i.e., matching frequency with exogenous otDCS) and a potent marked power increase in the above bands (Figure 6).

**Table 1 brainsci-13-00072-t001:** Comparative study shows the results between presnet work and some previous works.

Study	Stimulation Patterns /Frequency	Study Samples	EEG Findings and Power Effect at Stimulation Frequency
Zaehle, et al. [45]	tACS (8–12 Hz)	Healthy participant	tACS to modulate ongoing brain waves
Binder et al. [46]	Slow otDCS (1.5 Hz)	Male Long Evans rats	Enhanced ongoing wave’s upperDelta (2–4 Hz) Drop in theta band
Helfrich et al. [47]	10 Hz (α-tACS)	Healthy subjects	Enhance the power of ongoing brain waves
Neuling et al. [48]	10Hz (α-tDCS)	Healthy subjects	Enhancement of α-power
Krause et al. [49]	20 Hz β-tACS	Healthy subjects	No effect

In this study, we have dealt with the capacities of both otDCS and tDCS during different stimulation patterns to modulate dominant endogenous oscillations. Looking forward, the present findings have to be compared with related oscillated TES, such as tACS. At the same time, we have to keep in mind that oscillated current has specific behavioral interactions with endogenous brain rhythms that are distinguishable from other oscillated current. In one study, they revealed that the application of tACS with exogenous stimulation modulates those frequency-matching endogenous oscillations [50]. In addition, another previous study demonstrated that stimulation with alpha-10 tACS entrains individual intrinsic alpha-band ~10 ± 2 Hz [50].

In Group B, which was characterized with otDC stimulation followed by a tDC stimulation, it was found that tDCS was acting as a power generator (Figure 8), in which after otDCS at 8.5 Hz, it made a mild increase in alpha-1B power (*p*: 0.81, F (2, 14)3.7: 00) (Figure 5). Furthermore, the results showed a marked increase in the power of the upper bands in an ascending manner: alpha-2B (*p* < 0.01, F (2, 14)3.7: 21), beta-1B (*p* < 0.01, F (2, 14)3.7: 10.2), and beta-2B (*p* < 0.05, F (2, 14)3.7: 4.9) (Figure 5). The previous group, Group B, exhibited a lower potent power change in alpha-1B power corresponding with a change in alpha-1A power (F (2, 14)3.7: 4.7, *p* < 0.05) (Figure 5). On the other hand, the upper bands of Group B revealed a greater potent power increase than corresponding with bands in Group A: alpha-2A (*p* < 0.05, F (2, 14)3.7: 5.6), beta-1A (F (2, 14)3.7: 5.02, *p* < 0.05), and beta-2A (F (2, 14)3.7: 2.2, *p*: 0.15) (Figure 5). In the Group C pattern, the behavior of the exogenous otDCS/tDCS was like that of the Group B pattern but with a lower potent effect. In it, otDCS at 13 Hz followed by tDCS had a mild capacity to increase the power of beta-1C (F (2, 14)3.7 = 0.31, *p*: 0.7) (Figure 5) and more with beta-2C (F (2, 14)3.7: 0.14 and *p*: 0.71 (Figure 5)). In fact, all these findings simulated some revealed aspects of both tDCS and oscillated TES, in which anodal tDCS (not oscillated TES) has the ability to increase the power and firing rate, while oscillated TES is capable of regulating the firing rate up and down the targeted intrinsic oscillation [51]. Thus, clearly, one can observe the ability of the applied otDCS to regulate an increase in power generation by tDCS in the upper intrinsic bands.

In fact, reviewing the changes in alpha-1A power and alpha-2A power (Figure 7) revealed an increase in the cumulative band power changes by tDCS-otDCS at 8.5 Hz and tDCS-otDCS at 13 Hz, respectively. The linking of these results with those of the first-session power changes (dP) by otDCS at 8.5 Hz and otDCS at 13 Hz (Figure 8) demonstrated that the tDCS-otDCS stimulation patterns have the ability to increase the power of the matched intrinsic oscillations with a more potent power effect in the upper bands. Thus, one can conclude that stimulation with otDCS and tDCS patterns can increase the power of those matched endogenous bands with a more potent effect in the upper bands. More precisely, the findings revealed that the increase in the previous bands’ power (frequency matching and upper bands) was more markedly potent by the otDCS-tDCS stimulation patterns than by the tDCS-otDCS patterns.

## 5. Conclusions

In summary one can conclude that:Well-known anodal tDCS has a general acute potent stimulation power, which is non-specific for matched endogenous oscillation bands.Stimulation of the brain cortex with anodal otDCS makes the mild power effect on frequency-matched endogenous bands acute. However, if it is followed by anodal tDCS, it will make a marked acute power increase in the matched endogenous band.There was evidence that anodal otDCS with a certain frequency rate followed by anodal tDCS produces a marked acute power increase in the above bands of the matched endogenous band. However, the same result did not appear in the case of the anodal tDCS-otDCS pattern (i.e., starting with anodal tDCS followed by anodal otDCS within stimulation patterns).The findings suggested that the otDCS-tDCS pattern results in a mild increase in the power of the matched endogenous band (analogs of otDCS frequency) and a marked increase in the upper bands in an ascending manner.

## 6. Recommendations

Further experiments have to be performed to measure the minimum number of stimulated sessions of both otDCS (as band guidance) and tDCS (as power generator), according to the otDCS-tDCS pattern to make the desired power band change. Additionally, further studies should be conducted to check the terminal upper limit of power changes during this pattern. This can help us suggest a complete stimulation protocol to stimulate the desired brain band using the otDCS–tDCS stimulation pattern protocol.

## Figures and Tables

**Figure 1 brainsci-13-00072-f001:**
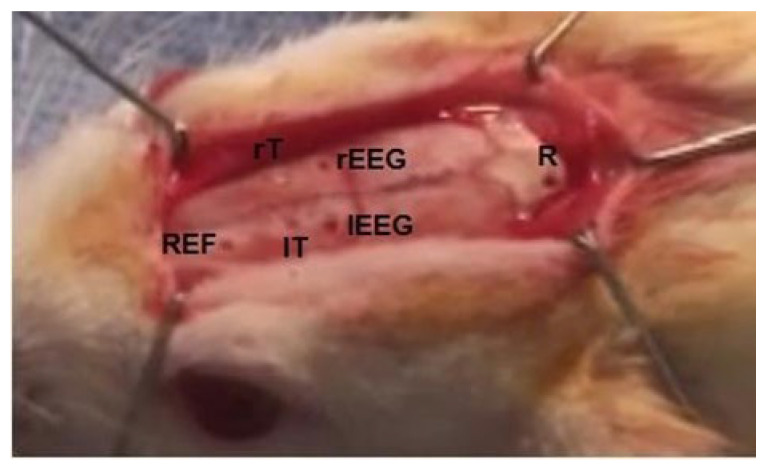
Making of transcranial and epidural holes over the dorsal surface of rat skull for both TES and EEG of FC, respectively. rT and lT; are right and left transcranial holes for TES, respectively. R; a hole for implantation of an electrode to be as return electrode for TES, and ground electrode for epidural EEG. rEEG and lEEG; both for right and left epidural EEG recording, while REF; for reference electrode.

**Figure 2 brainsci-13-00072-f002:**
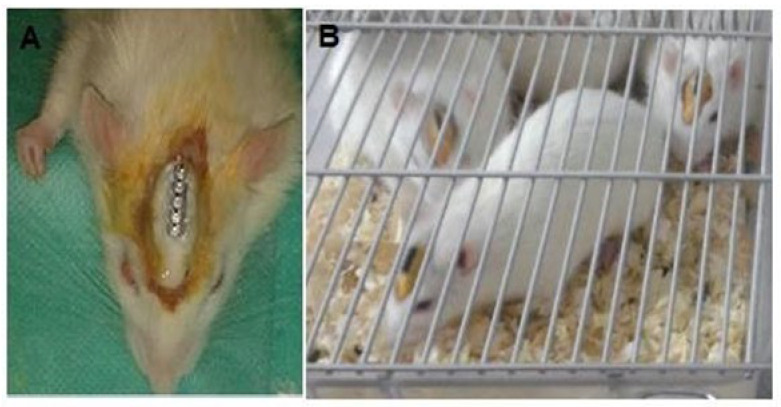
Fixation of 5-pin male connector tightly with the skull by hump-shaped dental cement. (**A**) Rat with implanted 5-pin male connector after surgery. (**B**) Rats with implanted 5-pin male connectors one week after surgery.

**Figure 3 brainsci-13-00072-f003:**
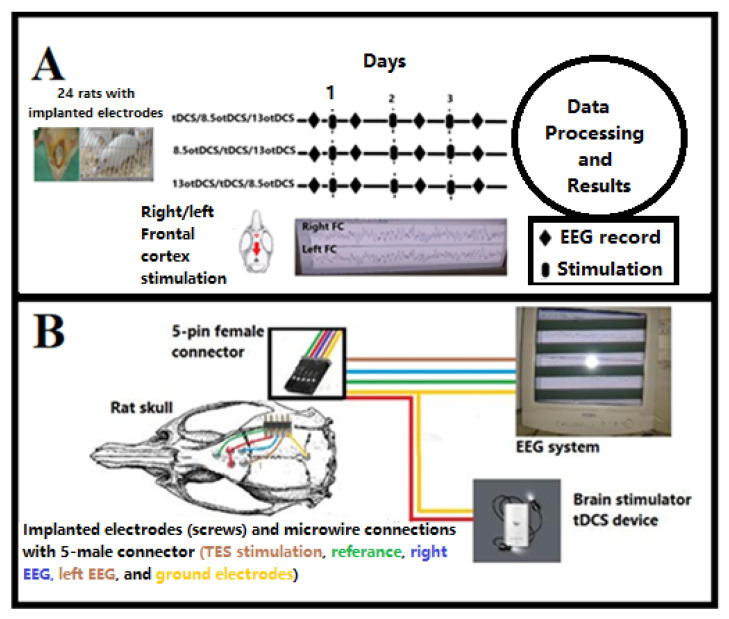
(**A**) Experimental protocol and (**B**) schematic diagram of present study. Schematic diagram views micro-wire connections between implanted electrodes and both EEG system and electrical stimulator device, via 5-pin male connector. (Note that yellow wire connecting with occipital electrode act as ground electrodes for epidural EEG and return electrode for anodal tDCS/otDCS).

**Figure 4 brainsci-13-00072-f004:**
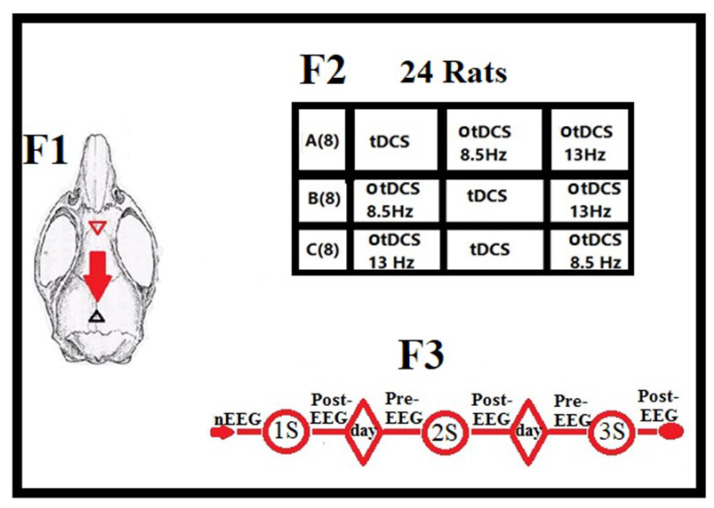
Details about current direction of anodal tDCS (red triangle) over FC toward occipital cathode (black triangle), stimulation patterns of three animal groups, and time-line plan for stimulation series in each group. F1: Targeted area (FC) was under anodal stimulation 0.2 mA/15–20 min for both tDCS and otDCS. F2: A total of 24 rats were investigated in this study, having been divided into 3 groups (A, B, and C; eight rats in each) with specific stimulation patterns of each. F3: Sessions of stimulation with time intervals for each group; at first, normal EEG (nEEG) record was harvested, following with applied first stimulation (1S) and after 10–20 min post-EEG was recorded. Second stimulation (2S) was carried out after 24 h, which was preceded by pre-EEG and followed with post EEG after 10–20 min. termination of stimulation sessions, third stimulation (3S) like that in (2S). Note: At least 5 min of recorded EEG were collected in each recording time period.

**Figure 5 brainsci-13-00072-f005:**
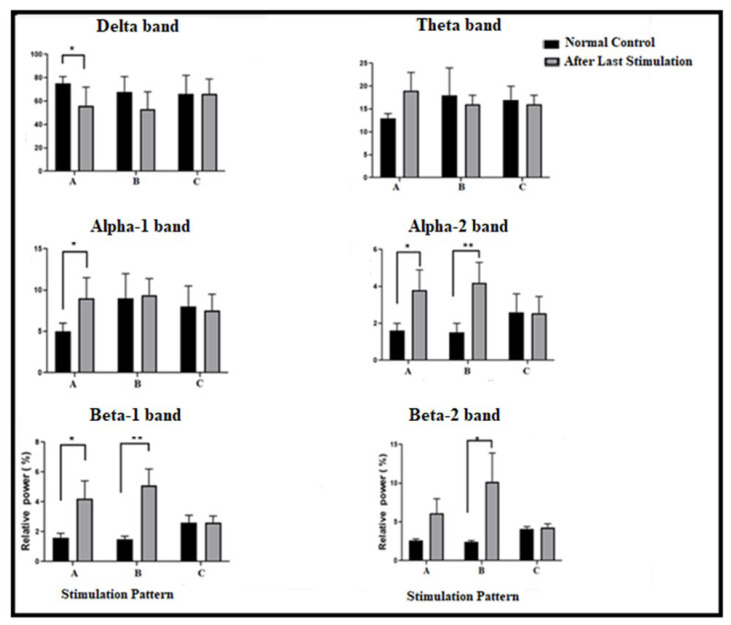
Topographical changes in the mean relative powers (±SEM) between normal and stimulation pattern effect (after last stimulation within group pattern) in all groups (A, B, and C), for delta band (1.5–4 Hz), theta band (4–7 Hz), alpha-1 band (7–10 Hz), alpha-2 band (10–12 Hz), beta-1 band (12–15 Hz), and beta-2 band (15–30 Hz). Data are represented as mean ± SEM, *n* = 8, and statistical analysis by one-way ANOVA followed by Tukey’s post hoc test. * *p* < 0.05 and ** *p* < 0.01 considered a significant result.

**Figure 6 brainsci-13-00072-f006:**
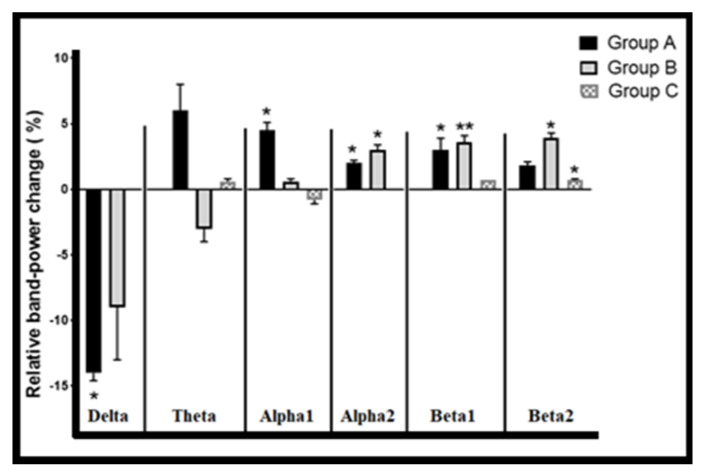
Topographical power changes between normal (baseline; before starting of stimulation pattern) and acute power change after last stimulation (i.e., pattern effect) in three stimulation patterns (i.e., power increase: above Y-0 axis, decreasing; under Y-0 axis and no change equal Y-0 axis). A: Group A stimulation patterns. B: Group B stimulation patterns. C: Group C stimulation pattern. Delta: 1.5–4 Hz, theta: 4–7, alpha-1: 7–10, alpha-2: 10–12, beta-1: 12–15, beta-2: 15–30. Mean relative powers (±SEM) between normal and stimulation pattern effect (after last stimulation within group pattern) in all groups and baseline level (before starting) of stimulation pattern. Data are represented as mean ± SEM, *n* = 8, and statistical analysis by one-way ANOVA followed by Tukey’s post hoc test. * *p* < 0.05 and ** *p* < 0.01 considered a significant result.

**Figure 7 brainsci-13-00072-f007:**
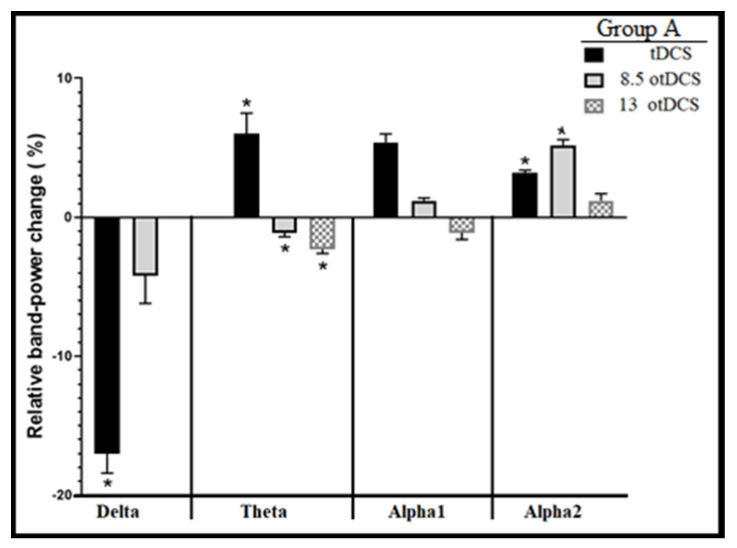
Four bands (delta, theta, alpha-1, and alpha-2) with significant cumulative changes in power differences (dP) induced by Group A stimulation pattern. Mean relative powers (±SEM) between normal and stimulation pattern effect by Group A stimulation pattern (*, *p* < 0.05). Data are represented as mean ± SEM, *n* = 8 and, statistical analysis by one-way ANOVA followed by Tukey’s post hoc test. * *p* < 0.05 considered a significant result.

**Figure 8 brainsci-13-00072-f008:**
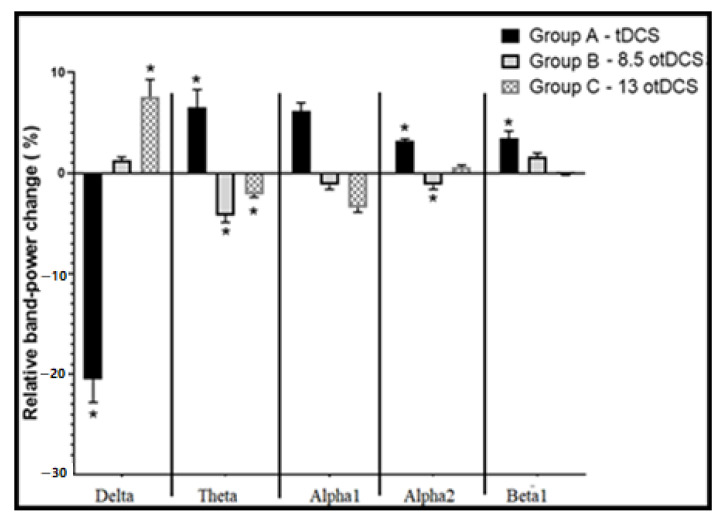
Acute Mean relative powers (±SEM) in the relative powers (dP) in five investigated bands were stimulated by first stimulation session by Group A: tDCS, Group B: 8.5 Hz otDCS, Group C: 13 Hz otDCS. Mean relative powers (±SEM) between normal and stimulation pattern effect. Data are represented as mean ± SEM, *n* = 8, and statistical analysis by one-way ANOVA followed by Tukey’s post hoc test. * *p* < 0.05 considered a significant result.

## Data Availability

This work has been done by Nafe and his colleagues. All EEG data are archived in hard disk and it available from the corresponding author under request.

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
