# Peer review of "Effectiveness of Anodal otDCS Following with Anodal tDCS Rather than tDCS Alone for Increasing of Relative Power of Intrinsic Matched EEG Bands in Rat Brains"

_brainsci, 2022, doi:10.3390/brainsci13010072_

Round 1
Reviewer 1 Report
The manuscript presents experimental results on different combinations of tDCS and otDCS at different frequencies. The scientific questions raised are pertinent and timely and the statistical evaluation is reasonable. The work is interesting and promises to add to the understanding of transcranial electric stimulation.
Some comments that should be addressed :
* page 2: “and all this can be related to otDCS (13) ”: it seems that (13) is not related to otDCS. Please clarify the usage of this reference. In addition, otDCS is not defined and this has to be done. Especially it is necessary to point out the difference to the established technique tACS. Introducing otDCS, it is important to clarify why this is called “anodal oscillatory tDCS”: if the electric current oscillates about zero as in tACS, it is not reasonable to call it “anodal” due to the polarity changes. Hence a clear explanation on the polarity changes is mandatory.
* page 2: “Therefore, unlike tDCS, the application of otDCS can be done non-invasively”: this seems to say that tDCS is applied invasively, which is not correct. tDCS is implies non-invasive neurostimulation.
* page 2: “During TES studies, ”: define TES
* page 4: “((2))”: typo
* page 4: “Liebetanz and his colleagues, 2009 have ”: typo
* page 4: “habituation of the rats to a restricted area to remain quiet ”: which restricted area ? In the cage ? Please clarify.
* caption Fig.4: “at least 5min of EEG recorded have collected in each recording time”: rephrase, maybe “at least 5min of recorded EEG have been collected in each recording time period”.
* page 6: “recorded pre- post stimulation”: rephrase, like “recorded pre- and post stimulation ”
* page 6, section 2.7, “5min of EEG recorded have collected in each recording time ”: rephrase, see above.
* page 6, section 2.7, “is not capable of analyzing such types of data, iWorx LabScribe 3 can. Therefore, all stored EEG data was transferred to the iWorx LabScribe3 system. ”: re-phrase
* page 7, “The main source of external electrical wave from amplify EEG, tDCS stimula- tion, and cables. ”: rephrase
* page 7, “Copyright 1984- 2011, MathWorks, Inc., USA, 2011 ”: remove, too much information
* page 7, “each EEG_DATA artifact-free ” : what is EEG_DATA ? Define or remove it.
* page 7, “relative band power (RBP) ”: please define, this is spectral power relative to what ?
* caption of Fig.7, “stimulation pattern effect in by group A stimulation pattern ”: re-phrase. In addition, it seems that two different text fonts/text sizes have been used in the cpation, please check.
* page 10, “the same order stimulation within all group patterns ”: ??? what is an “order stimulation”, do you mean “stimulation order” ? Please clarify. Similarly, what are “most first-order stimulation patterns ” ? Please add explanations, otherwise the respective Figure 8 is not clear at all.
* caption of Table 1, “presnet”: typo
* Discussion: no results should be given in the Discussion or re-iterated in detail, e.g. on page 13. Provide these in the Results section or remove them.
* Table 1 is not referred to in the text. Please include its content in the text or remove it.
* section 5, “one can concluded that ”: “one can conclude that”
Author Response
Reviewer 1 noes
- page 2: “and all this can be related to otDCS (13) ”: it seems that (13) is not related to otDCS. Please clarify the usage of this reference. In addition, otDCS is not defined and this has to be done. Especially it is necessary to point out the difference to the established technique tACS. Introducing otDCS, it is important to clarify why this is called “anodal oscillatory tDCS”: if the electric current oscillates about zero as in tACS, it is not reasonable to call it “anodal” due to the polarity changes. Hence a clear explanation on the polarity changes is mandatory.
OK, REQUESETED CHANGES HAVE BEEN DONE, otDCS is not oscillated under/above the zero like ACS, current oscillated around a certain DC value above the zero (anodal) , so could be with two polarity anodal/cathode, you can see this reference:
where current is oscillating around a certain value (either positive or negative),
D'Atri, A., De Simoni, E., Gorgoni, M., Ferrara, M., Ferlazzo, F., Rossini, P. M., & De Gennaro, L. (2015). Frequency-dependent effects of oscillatory-tDCS on EEG oscillations: A study with better oscillation detection method (BOSC). Archives Italiennes de Biologie, 153(2-3), 124-134.
Vulić, K., Bjekić, J., Paunović, D., Jovanović, M., Milanović, S., & Filipović, S. R. (2021). Theta-modulated oscillatory transcranial direct current stimulation over posterior parietal cortex improves associative memory. Scientific Reports, 11(1), 1-8.
Changes had done.
- page 2: “Therefore, unlike tDCS, the application of otDCS can be done non-invasively”: this seems to say that tDCS is applied invasively, which is not correct. tDCS is implies non-invasive neurostimulation.
Ok, done
- page 2: “During TES studies, ”: define TES
Ok done
- * page 4: “((2))”: typo
Ok done
- * page 4: “Liebetanz and his colleagues, 2009 have ”: typo
Ok done
- * page 4: “habituation of the rats to a restricted area to remain quiet ”: which restricted area ? In the cage ? Please clarify.
Answer just in the next sentence ( closed box ….etc)
- * caption Fig.4: “at least 5min of EEG recorded have collected in each recording time”: rephrase, maybe “at least 5min of recorded EEG have been collected in each recording time period”.
Ok, done thanks J
- * page 6: “recorded pre- post stimulation”: rephrase, like “recorded pre- and post stimulation ”
Ok done
- * page 6, section 2.7, “5min of EEG recorded have collected in each recording time ”: rephrase, see above.
Ok done
- * page 6, section 2.7, “is not capable of analyzing such types of data, iWorx LabScribe 3 can. Therefore, all stored EEG data was transferred to the iWorx LabScribe3 system. ”: re-phrase
Ok done
- * page 7, “The main source of external electrical wave from amplify EEG, tDCS stimula- tion, and cables. ”: rephrase
Ok done
- * page 7, “Copyright 1984- 2011, MathWorks, Inc., USA, 2011 ”: remove, too much information
Ok done
- * page 7, “each EEG_DATA artifact-free ” : what is EEG_DATA ? Define or remove it.
Ok done
- * page 7, “relative band power (RBP) ”: please define, this is spectral power relative to what ?
Relative band power is well-known as the fraction/ratio of absolute power of wither delta or theta ..etc to the overall total powers of all captured bands expressed in % . any way I will add it.
Ok done
- * caption of Fig.7, “stimulation pattern effect in by group A stimulation pattern ”: re-phrase. In addition, it seems that two different text fonts/text sizes have been used in the cpation, please check.
Ok done
- * page 10, “the same order stimulation within all group patterns ”: ??? what is an “order stimulation”, do you mean “stimulation order” ? Please clarify. Similarly, what are “most first-order stimulation patterns ” ? Please add explanations, otherwise the respective Figure 8 is not clear at all.
Ok done changed all stimulation order ( 1,2,3) to stimulation sessions . I think now it is clear
- * caption of Table 1, “presnet”: typo
Ok done
- * Discussion: no results should be given in the Discussion or re-iterated in detail, e.g. on page 13. Provide these in the Results section or remove them.
Ok done , removed
- * Table 1 is not referred to in the text. Please include its content in the text or remove it.
Ok done
- * section 5, “one can concluded that ”: “one can conclude that”
Ok done

Reviewer 2 Report
The study by Al-Tawarah et al. evaluates the modulatory after-effects on EEG activity of three different sequences of transcranial electrical stimulation where tDCS and otDCS (at two frequencies) are alternated.
The study aims are of great interest for identifying novel effective protocols of tES in the clinical settings.
Nevertheless, the manuscript is not well written and the research design has several weaknesses. In addition, results are really divergent from the current literature which has not been sufficiently scrutinized and discussed in the paper.
Major points:
- since the study is based on after effects (not acute effects during stimulation), it should be particularly important to provide exemplary EEG traces and behavioral analysis of the animal's activity during recordings. As the matter of facts, some important changes in EEG bands might be related to changes in behavior, e.g. sleep.
- the sequence of stimuli is produced across 3 days, which renders the results of pre-post comparisons even more questionable (see also previous note). Also the after effects cannot be ascribed to the whole stimulation pattern, rather to the final session (and the results are questionable, see below). The reason for this choice is not provided.
- most current literature on animal studies did not provide evidence for enduring modulatory effects on tACS / otDCS, i.e. after-effects (this should be better stated and discussed). Thus, the after-effects here reported relative to sessions where only otDCS is produced (e.g. B and C in Figure 8) should be better validated by providing behavioral data, EEG traces and single values for each group in bar plot.
- otDCS is not equivalent to tACS, and this difference should be better described and referenced by authors, while the references provided are not related to this aspect.
Minor points:
- figures describing the experimental design/setup are of bad quality and not clear enough.
- Description of the experimental phases and analyses should be more clear and easy to understand.
- The language is very poor and most sentences of the paper must be rephrased (including title). Also use of some words is confounding (e.g. normal control for eeg baseline)
Author Response
Reviewer 2 notes and point by point response
- since the study is based on after effects (not acute effects during stimulation), it should be particularly important to provide exemplary EEG traces and behavioral analysis of the animal's activity during recordings. As the matter of facts, some important changes in EEG bands might be related to changes in behavior, e.g. sleep.
In fact, EEG data was analyzed for the same animal before/after stimulation so state was the same for the animal and just any change in EEG traces supposed to be related for stimulation session. So, one thing should be excluded is just internal and external stimuli (noise/ artifact) that we did.
- the sequence of stimuli is produced across 3 days, which renders the results of pre-post comparisons even more questionable (see also previous note). Also the after effects cannot be ascribed to the whole stimulation pattern, rather to the final session (and the results are questionable, see below). The reason for this choice is not provided.
Just notifiable marked EEG changes were discussed during the results/discussion either if its marked or significant changes. Sessions in fact could be more than 3 days but we just want to understand here if we stimulate the brain with the same ongoing rats’ brain EEG (related for state) following/before well-known anodal DC, is the brain EEG response is the same as if just stimulating the brain with DC in all sessions. This work pointed to a marked and some significant changes in different patterns, work need further behavioral studies and neuro-chemical consideration…etc.
- most current literature on animal studies did not provide evidence for enduring modulatory effects on tACS / otDCS, i.e. after-effects (this should be better stated and discussed). Thus, the after-effects here reported relative to sessions where only otDCS is produced (e.g. B and C in Figure 8) should be better validated by providing behavioral data, EEG traces and single values for each group in bar plot.
I pointer for some studies, as well as this work I think it is new pointed to successful cooperation between otDCS and DC rather than application of DC alone during anodal-tES to increase the power of targeted brain region. I think this work will increase the performance and effectiveness of tES, in which hybridization of otDC/DC more effective than tDCS alone.
- otDCS is not equivalent to tACS, and this difference should be better described and referenced by authors, while the references provided are not related to this aspect.
OK, otDCS is not oscillated under/above the zero like ACS, current oscillated around a certain DC value above the zero (anodal) , so could be with two polarity anodal/cathode, you can see this reference:
where current is oscillating around a certain value (either positive or negative),
D'Atri, A., De Simoni, E., Gorgoni, M., Ferrara, M., Ferlazzo, F., Rossini, P. M., & De Gennaro, L. (2015). Frequency-dependent effects of oscillatory-tDCS on EEG oscillations: A study with better oscillation detection method (BOSC). Archives Italiennes de Biologie, 153(2-3), 124-134.
Vulić, K., Bjekić, J., Paunović, D., Jovanović, M., Milanović, S., & Filipović, S. R. (2021). Theta-modulated oscillatory transcranial direct current stimulation over posterior parietal cortex improves associative memory. Scientific Reports, 11(1), 1-8.
Changes had done.
Minor points:
- figures describing the experimental design/setup are of bad quality and not clear enough.
ok
- Description of the experimental phases and analyses should be more clear and easy to understand.
Ok
- The language is very poor and most sentences of the paper must be rephrased (including title). Also use of some words is confounding (e.g. normal control for eeg baseline)
Ok I did, I sent editing and proofread certificate
Round 2
Reviewer 1 Report
My comments have been addressed sufficiently.